# Clinical transplantation using negative pressure ventilation ex situ lung perfusion with extended criteria donor lungs

Max T. Buchko [1,2], Nasim Boroumand[1,2], Jeffrey C. Cheng[1,2], Alim Hirji [3,4,5], Kieran Halloran[3,4,5], Darren H. Freed [1,2,4,5] & Jayan Nagendran [1,2,4,5 ✉]

Lung transplantation remains the best treatment option for end-stage lung disease; however, is limited by a shortage of donor grafts. Ex situ lung perfusion, also known as ex vivo lung perfusion, has been shown to allow for the safe evaluation and reconditioning of extended criteria donor lungs, increasing donor utilization. Negative pressure ventilation ex situ lung perfusion has been shown, preclinically, to result in less ventilator-induced lung injury than positive pressure ventilation. Here we demonstrate that, in a single-arm interventional study (ClinicalTrials.gov number NCT03293043) of 12 extended criteria donor human lungs, negative pressure ventilation ex situ lung perfusion allows for preservation and evaluation of donor lungs with all grafts and patients surviving to 30 days and recovered to discharge from hospital. This trial also demonstrates that ex situ lung perfusion is safe and feasible with no patients demonstrating primary graft dysfunction scores grade 3 at 72 h or requiring post-operative extracorporeal membrane oxygenation.

[1] Division of Cardiac Surgery, Department of Surgery, University of Alberta, Edmonton, AB, Canada. [2] Mazankowski Alberta Heart Institute, Edmonton, AB, Canada. [3] Division of Pulmonary Medicine, Department of Medicine, University of Alberta, Edmonton, AB, Canada. [4] Alberta Transplant Institute, Edmonton, AB, Canada. [5] Canadian Donation and Transplantation Research Program, Edmonton, AB, Canada. ✉email: jayan@ualberta.ca

Pulmonary transplantation remains the best long-term therapeutic option for the management of end-stage lung disease. Despite increasing rates of lung transplantation, primarily through an increased use of donation after circulatory death donors (DCD), transplant waitlist mortality has traditionally been relatively static. For example, in the United States, average waitlist mortality has been consistently between 15 and 18 per 100 waitlist years over the last decade[1]. This is likely to increase acutely; however, due to a significant number of patients with SARS-CoV-2 induced pulmonary fibrosis[2]. Compounding the shortage of this life-saving resource is the fact that organ utilization rates of available grafts remains low (20–30%)[3–5]. Concerns regarding organ quality remain the most common reason for declining an organ offer[3].

The traditional method of organ preservation, cold static preservation (CSP), has several key limitations that limit its use to standard criteria donor lungs. During CSP, lungs are flushed with a low-potassium, dextran containing solution and transported on ice back to the center of implantation. Unfortunately, this method limits the preservation time to less than six hours; after which, rates of primary graft dysfunction (PGD) and subsequent mortality rise exponentially[6]. Furthermore, while static, the organs cannot be reassessed prior to implantation. Finally, there is limited potential for delivering therapeutic agents as there is neither circulatory, nor ventilatory flow to the lungs.

Ex situ lung perfusion (ESLP) is a developing technology that has been used to evaluate and recondition marginal donor lungs and has repeatedly allowed for the utilization of extended criteria donor (ECD) lungs for transplantation with acceptable outcomes[7–11]. During ESLP, the lungs are continually perfused with a physiologic substrate at normothermic temperatures. Therapeutic agents may be added to the perfusate to optimize the function of extended criteria donors. They are also ventilated, allowing the assessment of organ function outside of the inflammatory milieu of the whole donor body. By doing so, utilization rates have increased by as much as 15–20% in some centres[7,12].

Currently, all clinically available ESLP devices utilize positive pressure ventilation (PPV). Positive pressure ventilation has the unintended consequences of distributing ventilatory pressure heterogeneously throughout the lung[13–15]. Negative pressure ventilation equally distributes the forces of inspiration homogeneously over the entire pleural surface the lung[16]. Negative pressure ventilation ex situ lung perfusion (NPV-ESLP) has been described pre-clinically to be associated with decreased ventilator-induced lung injury compared to PPV-ESLP[17].

Our objective was to assess the safety and efficacy of using NPV-ESLP in clinical transplantation of extended criteria donors. Extended criteria donors were procured using standard protocols for retrieval and CSP, then transported back to the site of implantation for NVP-ESLP. Lungs were then connected to the NPV-ESLP device for assessment, reconditioning and functional evaluation. If the organs were deemed suitable for transplantation, the organs were maintained on the device until the first recipient lung had been explanted; after which, they were flushed again with low-potassium dextran solution and transferred to the surgical field for implantation. Recipients were approached to participate in the study at the time of listing, gave informed consent and received routine institutional practices regarding post-operative care.

Clinical endpoints were serially monitored, including PGD scores at 0, 24, 48, and 72 h, time to extubation, length of ICU and hospital stay, requirements for extracorporeal membrane oxygenation, and graft and patient survival.

## Results

**Recruitment and donor characteristics**. Lungs were recruited from twelve extended criteria donors from October 2018 to July 2019. All lungs were subsequently perfused using the NPV-ESLP device. All twelve lungs perfused on the device were deemed suitable for transplantation and subsequently implanted.

Donor characteristics are described in Table 1. Nine out of twelve of the donors followed donation after neurologic determination of death (NDD). The average pre-procurement donor P:F ratio was $234 \pm 38$ mmHg. In NDD donors, the pre-procurement P:F ratio was $174 \pm 24$ mmHg. The average Eurotransplant ECD donor score was $9.6 \pm 0.6$ (Supplementary Table 2)[18].

**Preservation details**. The mean total cold ischemic time was $308 \pm 27$ min for the left lung and $359 \pm 25$ min for the right lung. The average time on ESLP was $182 \pm 11$ min. The average total time from donor explant to re-implantation was 8 h 14 min $\pm$ 33 min for the left lung and 9 h 6 min $\pm$ 31 min for the right lung. Graft ischemic and ESLP times are summarized in Supplementary Table 3. The mean final P:F ratio on the NPV-ESLP device was $492 \pm 30$ mmHg. There were no organs that did not meet the criteria for utilization, following NPV-ESLP. Criteria for utilization had been met by 2 h of ESLP in 7/12 cases. The remaining five lungs had demonstrated sufficient and stable hemodynamic parameters and oxygenation over the first 3 h and were subsequently accepted following the 3rd hour evaluation period.

Dynamic lung compliance, pulmonary artery pressures, and transpulmonary pressure ($P_{L} = P_{aw} - P_{pleural}$, difference between airway pressure and pleural pressure) are demonstrated in Fig. 1. All three parameters demonstrated a relatively linear trend during the period of ESLP in all lungs. Overall transpulmonary pressures were relatively low. For example, the average transpulmonary pressure within the chamber was $13.8 \pm 0.6$ cmH$_2$O amongst all lungs at T2.

**Recipient outcomes**. Recipient outcomes are summarized in Table 2. The primary endpoint, including survival to 30 days post-transplant and absence of primary graft dysfunction grade 3 (PGD3) within 72 h after transplantation was met in all recipients. Furthermore, all grafts and patients recovered to discharge from hospital and survived to 1-year post-transplantation. The

### Table 1 Donor demographic details*.

| Donor characteristics | Trial patients ($n = 12$) |
|---|---|
| Age (yrs) | 43 ± 3 |
| Height (m) | 1.69 ± 0.02 |
| Weight (kg) | 97.5 ± 9.1 |
| Cause of death—*n* (%) | |
| Intracranial bleed | 5 (42) |
| Overdose | 3 (25) |
| CVA | 1 (8) |
| Gunshot wound to head | 1 (8) |
| Hypoxic brain injury from aspiration | 1 (8) |
| Bulbar ALS—MAID | 1 (8) |
| Donor classification—*n* (%) | |
| NDD | 9 (75) |
| DCD | 3 (25) |
| *Donor P:F ratio (mmHg)* | |
| Total | 234 ± 38 |
| NDD | 174 ± 24 |
| DCD | 415 ± 46 |
| Eurotransplant ECD donor score | 9.6 ± 0.6 |

*ECD* extended criteria donor, *NDD* neurologic determination of death, *DCD* donation after circulatory death, *P:F Ratio* Ratio of PaO2:fraction of inspired oxygen, *CVA* cerebral vascular accident, *MAID* medical assistance in dying, *ALS* amyotrophic lateral sclerosis.
*Plus–minus values represent mean ± SE.

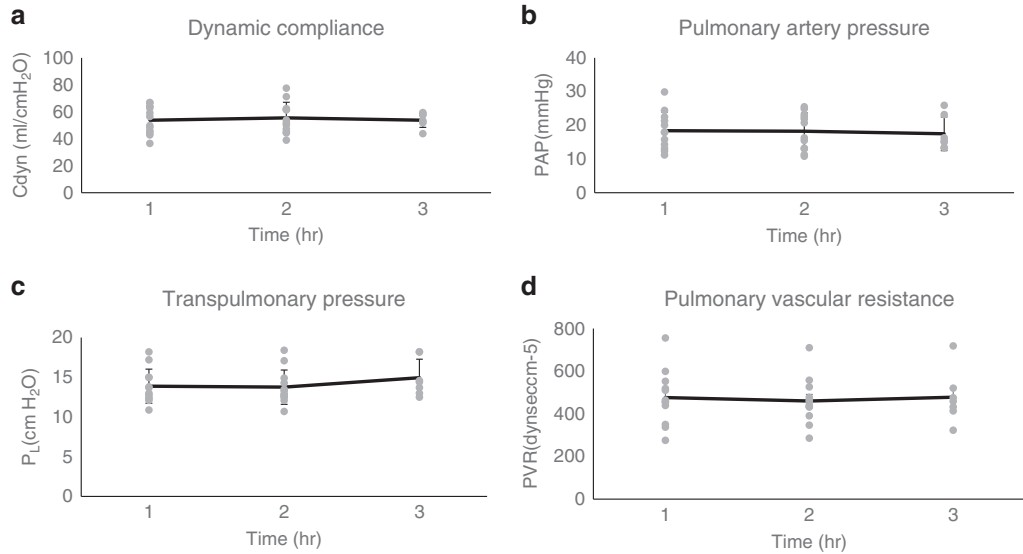

**Fig. 1 NPV-ESLP demonstrated stable hemodynamic parameters over the course of perfusion.** Dynamic compliance (Cdyn) (**a**), pulmonary artery pressure (PAP) (**b**), transpulmonary pressure ($P_L$) (**c**), and pulmonary vascular resistance (PVR) (**d**) over time. Results are expressed as mean ± SE ($n_{T1}$ = 12, $n_{T2}$ = 12, $n_{T3}$ = 5).

| Table 2 Recipient demographic details and clinical outcomes*. | |
| --- | --- |
| **Donor characteristics** | **Trial patients ($n = 12$)** |
| Age (yrs) | 58 ± 3 |
| Height (m) | 1.70 ± 0.02 |
| Weight (kg) | 76.1 ± 3.9 |
| Indication for transplantation—$n$ (%) | |
| Emphysema | 5 (42) |
| Talcosis | 2 (17) |
| IPF | 2 (17) |
| A1ATD | 1 (8) |
| Cystic fibrosis | 1 (8) |
| NSIP | 1 (8) |
| Mechanical ventilation (hours) | 30 ± 6 |
| ICU LOS (days) | 5.3 ± 0.7 |
| Post-transplant ECMO | 0 |
| PGD at 24 h—$n$ (%) | |
| Grade 0 | 9 (75) |
| Grade 1 | 0 |
| Grade 2 | 3 (25) |
| Grade 3 | 0 |
| PGD at 72 h—$n$ (%) | |
| Grade 0 | 10 (83) |
| Grade 1 | 0 |
| Grade 2 | 2 (17) |
| Grade 3 | 0 |

*ECMO* extracorporeal mechanical oxygenation, *ICU* intensive care unit, *LOS* length of stay, *PGD* primary graft dysfunction, *COPD* chronic obstructive pulmonary disease, *IPF* idiopathic pulmonary fibrosis, *NSIP* nonspecific interstitial pneumonitis, *ILD* interstitial lung disease, *A1ATD* alpha-1 anti-trypsin deficiency.
*Plus–minus values represent mean ± SE.

average recipient P:F ratio at 72 h post-transplant was 335 ± 41 mmHg. The average duration of mechanical ventilation was 30 ± 6 h. The average ICU length of stay was 5.3 ± 0.7 days. The average index hospital length of stay was 34.8 ± 7.4 days. The average FEV1 at 30 days was 2.3 ± 0.4 L. These outcomes are consistent with a contemporary cohort of recipients from our institution receiving standard criteria donor lungs without ESLP (Supplementary Table 4).

## Discussion

Machine perfusion has been regarded as one of the most important advancements in transplantation since the development of immunosuppression[19]. Optimized machine preservation has the potential to remove geographical constraints to transplantation, improve matching for disadvantaged patients, increase organ quality through organ-specific therapeutic interventions, improve disease screening, and induce immune tolerance[20]. Currently there are three commercially available ESLP systems: the XVIVO XPS, the OCS Lung, and the Vivoline LS1. An extensive review of the devices and their respective protocols has been described in detail by Sanchez et al.[21]. Most importantly, all protocols vary substantially with respect to several key perfusion and ventilation parameters, therefore leaving little consensus on the optimal methodology of ESLP. Significant research endeavors and advancements across multiple fields are needed before the potential of ESLP is fully realized.

This trial demonstrates the safety and feasibility of NPV-ESLP. All grafts evaluated ultimately were found suitable for utilization. Furthermore, satisfactory post-operative outcomes were achieved, including no primary graft dysfunction grade 3 at 72 h post-operatively.

Currently, ESLP is considered as a tool to assess and recondition marginal donor lungs. However, it must be recognized that when improperly implemented, it has the potential to damage otherwise suitable lungs. Extended criteria donor lungs have already sustained an acute lung injury prior to procurement; otherwise, they would exhibit normal function within the donor body. The introduction of technologies that better mimic the physiologic environment are ideally suited to safeguard against iatrogenic injury produced by machine perfusion. Furthermore, the development of more physiologic ex situ technologies may serve to create a more reliable model of predicting how the graft will perform post-transplantation.

ESLP has had poor clinical adoption despite proven efficacy in extended criteria donor lungs[4,22,23]. For example, in the US, between 2015 and 2018, only 5.9% of deceased donors underwent ESLP[24]. This lack of utilization is multifactorial, with key limitations including cost, lack of expertise, and lack of portability of certain devices. Another key limitation to the adoption of ESLP is a relatively low rate of organ utilization, following assessment and

reconditioning. For example, UNOS data indicates a discard rate of 43.4% of lungs perfused on EVLP[24]. This low rate of utilization is a key contributor to a lack of cost effectiveness of the technology[25]. Optimization of technologies for organ preservation can address the current hesitations surrounding machine perfusion and preservation.

Recent literature has demonstrated that the driving pressure of the respiratory system ($\Delta P = P_{PLAT} - PEEP$, difference between plateau pressure and peak end expiratory pressure) and the transpulmonary pressure ($P_L = P_{aw} - P_{pleural}$, difference between airway pressure and pleural pressure) are significant predictors of VILI[26]. These ventilation parameters also correlate with clinical outcomes[27]. Clinical guidelines have subsequently recommended that $P_{Plat}$ should be limited to $<30$ cmH$_2$O and $\Delta P < 25$ cmH$_2$O[28,29]. As previously stated, the average transpulmonary pressure within the chamber was $13.8 \pm 0.6$ cmH$_2$O amongst all lungs at T2. This corresponds to driving pressures of $8.8 \pm 0.6$ cmH$_2$O in the setting of negative pressure ventilation. These low ventilatory pressures corroborate our previous animal studies suggesting NPV provides safe and effective lung-protective ventilation during ESLP[17].

The goal of this trial is to demonstrate the safety and efficacy of the NPV-ESLP device and its ability to assess and reclaim marginal donor lungs. Therefore, the study was designed with the exclusion of recipients with pre-operative mechanical ventilation or mechanical circulatory support. Excluding these patients provided a degree of control over the contribution of recipient factors to post-transplant outcomes thereby giving a more focused assessment of the device-specific safety in this uncontrolled observational trial. As all donor grafts accepted for inclusion in the trial (up to a pre-specified 12 total transplants) were ultimately found to be suitable for transplantation after NPV-ESLP we were unable to exclude lungs based on the acceptance criteria of the trial. A larger scale, multi-center clinical trial is needed to move from safety assessment to determining overall efficacy before routine clinical adoption.

This trial demonstrates the safety and feasibility of NPV-ESLP. It also highlights that continual development of more physiologic systems of machine perfusion are needed if we are to realize the full potential of ex vivo organ perfusion.

## Methods

The study was approved by the institutional ethics review board at the University of Alberta [REB Approval: Pro00070552, (July 27, 2019)]. The study protocol is detailed in Supplementary Note 1. All participants gave written informed consent, according to CARE guidelines and in compliance with the Declaration of Helsinki principles. All transplants occurred at the University of Alberta, Edmonton, Alberta, Canada. Donor lungs were procured from throughout Western Canada and transported to the implantation site for ESLP and transplantation. Donor inclusion and exclusion criteria are summarized in Supplementary Table 1. Extended criteria donors included any of the following: P:F ratio less than 300 mmHg, Maastricht III or IV deceased from cardiac death donors (DCD), greater than 10 units of blood transfusion, expected cold ischemic time greater than 6 h, or donor age greater than 55 years old.

Donor lungs were flushed with 4 L of cold low-potassium dextran (LPD) solution (~10 °C) (Perfadex, Vitrolife, Gothenburg, Sweden) antegrade, followed by 1 L of retrograde flush divided between the pulmonary veins. Following procurement, the lungs were packaged in cold LPD solution and transported back to the site of implantation.

**Ex situ lung perfusion**. Following return to the University of Alberta, the lungs were immediately cannulated and connected to the NPV-ESLP device. The device perfusate contained STEEN solution (1.5 L) (XVIVO Perfusion, Gothenburg, Sweden), three units of Type-O red cell concentrate, Methylprednisolone (500 mg), unfractionated Heparin (40,000 units), Cefazolin (1000 mg), Voriconazole (200 mg), and Ciprofloxacin (400 mg). Insulin (Humulin R) was continually infused at 10IU/hr. Bolus doses of NaHCO$_3$ (8.4%) was intermittently given to maintain a physiologic pH.

Perfusion was initiated at 10% of predicted cardiac output (70 mL/kg) and a perfusate temperature of 21 °C. The perfusate temperature was gradually increased to achieve a final temperature of 37 °C within 15 min of perfusion. Once the

**Table 3 NPV-EVLP perfusion and ventilation strategy.**

|  | Preservation | Evaluation |
|---|---|---|
| Temperature | 37 °C | 37 °C |
| Pulmonary Artery Flow | 30% estimated CO (CO = 70 mL/kg/min.) | 50% estimated CO (CO = 70 mL/kg/min.) |
| *Ventilation parameters* |  |  |
| Mode | Volume control | Volume control |
| Desired inspiratory tidal volume | 8–10 mL/kg | 8–10 mL/kg |
| Frequency | 10 bpm | 10 bpm |
| Inspiratory: Expiratory Ratio | 1:1.5 | 1:1.5 |
| Peak TPGi | <21 cm H$_2$O | <25 cm H$_2$O |
| PEEP | 5 cm H$_2$O | 5 cm H$_2$O |
| FiO2 | 21% | 21% |
| *Perfusion parameters* |  |  |
| PAP | <30 mmHg | <30 mmHG |
| LAP | 0 mmHg | 0 mmHg |

*CO* cardiac output, *bpm* breaths per minute, *TPGi* inhaled transpulmonary gradient, *FiO₂* fraction inspired oxygen, *PEEP* positive end expiratory pressure, *PAP* pulmonary artery pressure, *LAP* left atrial pressure.

perfusate temperature had reached 32 °C, continuous positive airway pressure (CPAP) was applied at 15 cm H$_2$O. The trachea was then unclamped, preventing de-recruitment of the lungs. Ventilation was then initiated using a CPAP of 5 cm H$_2$O and end inspiratory pressure (EIP) titrated to achieve a tidal volume of 8–10 mL/kg of donor ideal body weight (Devine formula: IBW$_{men}$ = 50 kg + 2.3 kg × (height$_{(in)}$ − 60); IBW$_{women}$ = 45.5 kg + 2.3 kg × (height$_{(in)}$ − 60). Perfusate flow was increased in increments of 10% predicted cardiac output every 5 min until a flow of 30% cardiac output had been achieved.

Lung preservation was performed as described in Table 3. At 30 min, then hourly, an evaluation of lung performance was performed. A custom sweep gas (89% N$_2$, 8% CO$_2$, and 3% O$_2$) was applied at ~250 mL/min for 5 min to de-oxygenate the perfusate. Perfusate flow was simultaneously increased to 50% predicted cardiac output. Blood gases were taken at the end of the evaluation period. Settings were then returned to preservation mode, and the sweep gas was turned off.

**Organ evaluation**. Pulmonary artery pressure (PAP), pulmonary vascular resistance (PVR), dynamic compliance (Cdyn), were continually assessed over time. The ratio of partial pressure of oxygen in the pulmonary venous blood to the fraction of inspired oxygen (P:F Ratio) was also measured during evaluation.

**Acceptance and implantation**. Criteria for suitability for transplantation included maintenance of a P:F ratio > 300 mmHg and stable hemodynamic parameters (<10% decline in any of the following: PAP, PVR, and Cdyn) over the course of NPV-ESLP. Following acceptance of the organ, it was flushed with 3 L of cold low-potassium dextran (LPD) solution in an antegrade fashion, packaged on ice, and transferred to recipient OR. Standard implantation techniques were performed according to surgeon preference.

**Outcomes**. The primary endpoint was a composite of survival to 30 days post-transplant and absence of primary graft dysfunction grade 3 (PGD3) within 72 h after transplantation[30]. Secondary outcomes included PGD scores at 0, 24, 48, and 72 h post-transplantation, intensive care unit length of stay, hospital length of stay, duration of invasive mechanical ventilation post-transplantation, and survival to one-year post-transplantation.

**Data collection and statistics**. All results are expressed as mean ± standard error. Data were collected on Microsoft Excel Version 16.16.25 (Volume License 2016). All analyses were performed on STATA 15 (StataCorp LLC, College Station, Texas).

**Reporting summary**. Further information on research design is available in the Nature Research Reporting Summary linked to this article.

## Data availability

The data that support the findings of this study are available on request from the corresponding author, Jayan Nagendran. The data are not publicly available due to the University of Alberta Human Ethics Research Board patient confidentiality restrictions as they contain information that could compromise research participant privacy and consent. Source data are provided with this paper.

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

## Acknowledgements

This study was supported by grants from the Canadian Institutes for Health Research—Canadian Donation and Transplantation Research Program (CIHR-CDTRP), the University Hospital Foundation (UHF), and the Alberta Transplant Innovation Fund (ATIF).

## Author contributions

M.T.B. participated in research design, performance of the research, data analysis, and writing of the paper. J.C.C. participated in research design, performance of the research, and data analysis. N.B. participated in research design, performance of the research, and data analysis. A.H. participated in performance of the research and writing of the paper. K.H. participated in performance of the research, data analysis, and writing of the paper. D.H.F. participated in research design, performance of the research, data analysis, and writing of the paper. J.N. participated in research design, performance of the research, data analysis, and writing of the paper.

## Competing interests

Dr. Jayan Nagendran and Dr. Darren H. Freed are Co-Founders of TEVOSOL Inc. None of the remaining authors have any relationships with a commercial entity that has an interest in the subject of the presented manuscript or other conflicts of interest to disclose.
