## [Peer Review File · Nature Communications]

REVIEWER COMMENTS

Reviewer #1 (Remarks to the Author):

I thank to the authors and the editorial board for giving me this chance to review this article.

In this article Buchko et al. investigated their clinical experience utilizing NPV-ESLP for the assessment, preservation, and transplantation of 12 ECD human lungs. Their results are encouraging with 100% 30-day survival without PGD Grade 3 at T72.

I congratulate the team for their great effort.

Ex vivo lung perfusion or as the authors state ex situ lung perfusion is a great technology to test a questionable lung graft before implanting it. Although there is an accepted clinical practice (Toronto EVLP Protocol) in most of the transplant centers using this technology, there is still something to improve.

I think one of the ways for improvement is to apply physiologic condition such as “negative pressure” during assessment. This system is already in use for lung physiology assessments and air leak studies for stapler use.

1) The system that they use is a different than the wide used protocols, which makes it for me difficult to compare.

2) Another issue is the inclusion criteria: 6 of the 12 donors had P/F >200 mmHg which could have transplanted without any assessment.

3) 30-day survival: This is extremely short for assessment of success.

4) Although in the results mentioned that PVR is stable, there is no numerical value is given.

4) 3-h ESLP time is a little bit short compared to 4 to 6 h clinical application in other systems

5) I think there is writing failure in page 15: They do not start, I think, at 32°C with 10% flow. Is it 20°C?

Thanks again for giving me the opportunity to review this article.

Reviewer #2 (Remarks to the Author):

no comments

Reviewer #3 (Remarks to the Author):

This article is a report of a small (n=12) first in human pilot study of negative pressure ventilation (NPV) with extended criteria donor lungs. The statistics reported are descriptive statistics only. The study is well described and reported. There are no statistical issues. I note that the protocol states that this patient group will be compared with a contemporaneous group of patients receiving lungs according to the usual standard of care. This comparison is not reported. Neither is the longer term follow up at 6 months and one year. I conclude that this is the first report of this study and that a further report will follow.

Reviewer #1:

1) The system that they use is a different than the wide used protocols, which makes it for me difficult to compare.

Currently there are three commercially available ESLP systems. These are the XVIVO XPS system which uses the Toronto protocol, the Vivoline LS1 system which uses the Lund protocol, and the OCS Lung system which uses the OCS protocol. An extensive review of the systems and their respective protocols has been described in detail by Sanchez *et al*¹. In summary, all protocols vary substantially with respect to several key perfusion and ventilation parameters including pump type, ultimate flow rate, perfusate composition, target tidal volumes, deoxygenating gas mixtures, among others. With respect to ventilation, all systems currently available utilize positive pressure ventilation, whereas ours is the only device to utilize negative pressure ventilation. Currently, the OSC Lung system is the only portable system, and therefore is able to be transported to the site of procurement, whereas the XVIVO and Vivoline systems are not readily transported and require a period of cold static preservation and transport back to the site of implantation for assessment with ESLP. Contrastingly, the XVIVO system is the only system that utilizes acellular perfusates, which may have profound impact on edema formation and long-term preservation capabilities^{2,3}. Likewise, the Lund protocol is the only system that perfuses lungs at physiologic cardiac output, exerting significant hydrostatic pressure on the lungs, which also may have profound effects on long-term lung function and edema formation. To describe all the differences between our system and others and their predicted effects on successful perfusion would be the subject of a lengthy review article and is far outside the scope of the study. Furthermore, the intention of our study is not to compare outcomes with currently available systems, rather to determine the safety and efficacy of the device. Determining which ESLP system is superior would require a large, dedicated clinical trial to address the topic.

The discussion has been amended to:

Machine perfusion has been labelled as the “most important advancement in transplant technologies since the development of immunosuppression”[21]. Optimized machine preservation has the potential to remove geographical constraints to transplantation, improve matching for disadvantaged patients, increase organ quality through organ-specific therapeutic interventions, improve disease screening, and induce immune tolerance[22]. Currently there are three commercially available ESLP systems: the XVIVO XPS, the OCS Lung, and the Vivoline LS1. An extensive review of the devices and their respective protocols has been described in detail by Sanchez *et al* [23]. Most importantly, all protocols vary substantially with respect to several key perfusion and ventilation parameters, therefore leaving little consensus on the optimal methodology of ESLP. Significant research endeavours and advancements across multiple fields are needed before the potential of ESLP is fully realized.

2) Another issue is the inclusion criteria: 6 of the 12 donors had P/F >200 mmHg which could have transplanted without any assessment.

Donor inclusion and exclusion criteria are consistent with other major extended criteria donor EVLP trials including HELP trial⁴ by the Toronto group using the XVIVO device, the OSC EXPAND trial⁵ using the OCS Lung device. As such, the inclusion criteria were chosen to be consistent with the current body of literature.

3) 30-day survival: This is extremely short for assessment of success.

We appreciate 30 days survival is a relatively short assessment of success. We have amended the manuscript to also include 1-year survival which has recently concluded.

The primary goal of the trial is to determine whether the device has the ability to evaluate and recondition extended criteria donor lungs that may be transplanted with similar results to standard criteria donors. Whether ECD lungs have similar long-term outcomes as standard criteria donors is not within the scope of this trial. Furthermore, the long-term outcomes of lung transplant success are profoundly affected by other such as number of rejection episodes, frequency of infections, etc. That being said, multiple groups has published that long term EVLP outcomes in ECD donors are similar to standard criteria donors⁶⁻⁸.

4) Although in the results mentioned that PVR is stable, there is no numerical value is given.

The article has been amended to include PVR in Figure 1(d).

5) 3-h ESLP time is a little bit short compared to 4 to 6 h clinical application in other systems.

The purpose of the trial is to determine whether it may safely evaluate marginal donor lungs that would otherwise be rejected. Lungs were removed from the device and transplanted after they had met pre-specified criteria for utilization. This was generally by hour 3 on the device. No lungs perfused on the device needed more than 3 hours to reach the pre-specified criteria for utilization.

The trial was not designed to determine the safe storage duration of EVLP; however, our previously published work has shown normal function in porcine lungs on the device out to 24 hours^{9,10}.

6) I think there is writing failure in page 15: They do not start, I think, at 32°C with 10% flow. Is it 20°C?

This is correct. The article has been amended to read:

Perfusion was initiated at 10% of predicted cardiac output (70 mL/kg) and a perfusate temperature of 21°C. The perfusate temperature was gradually increased to achieve a final temperature of 37°C within 15 minutes of perfusion.

Reviewer #2: Reviewer #2 did not report any specific questions to address.

Reviewer #3:

- 1) I note that the protocol states that this patient group will be compared with a contemporaneous group of patients receiving lungs according to the usual standard of care. This comparison is not reported. Neither is the longer term follow up at 6 months and one year. I conclude that this is the first report of this study and that a further report will follow.**

This comment has been addressed in Response 3 to Reviewer #1. In addition, the manuscript has been amended to read:

The primary endpoint including survival to 30 days post-transplant and absence of primary graft dysfunction grade 3 (PGD3) within 72 hours after transplantation was met in all recipients. Furthermore, all grafts and patients recovered to discharge from hospital and survived to one-year post-transplantation. The average recipient P:F ratio at 72 hours post-transplant was 335 ± 41 mmHg. The average duration of mechanical ventilation was 30 ± 6 hours. The average ICU length of stay was 5.3 ± 0.7 days. The average index hospital length of stay was 34.8 ± 7.4 days. The average FEV1 at 30 days was 2.3 ± 0.4 L. These outcomes are consistent with a contemporary cohort of recipients from our institution receiving standard criteria donor lungs without ESLP (Table S5).

References:

1. Sanchez, P. G., Mackowick, K. M. & Kon, Z. N. Current state of ex-vivo lung perfusion. *Curr. Opin. Organ Transplant.* **21**, 258–266 (2016).
2. Nilsson, T. *et al.* Comparison of two strategies for ex vivo lung perfusion. *J. Hear. Lung Transplant.* **37**, 292–298 (2017).
3. Becker, S. *et al.* Evaluating acellular versus cellular perfusate composition during prolonged ex vivo lung perfusion after initial cold ischaemia for 24 hours. *Transpl. Int.* **29**, 88–97 (2016).
4. Cypel, M. *et al.* Normothermic Ex Vivo Lung Perfusion in Clinical Lung Transplantation. *N. Engl. J. Med.* **364**, 1431–1440 (2011).
5. Loor, G. *et al.* Portable normothermic ex-vivo lung perfusion, ventilation, and functional assessment with the Organ Care System on donor lung use for transplantation from extended-criteria donors (EXPAND): a single-arm, pivotal trial. *Lancet Respir.* **7**, 975–984 (2019).
6. Chakos, A., Ferret, P., Muston, B., Yan, T. D. & Tian, D. H. Ex-vivo lung perfusion versus standard protocol lung transplantation — mid-term survival and meta-analysis. **9**, 1–9 (2020).
7. Ghaidan, H. *et al.* Ten year follow-up of lung transplantations using initially rejected donor lungs after reconditioning using ex vivo lung perfusion. **1**, 1–8 (2019).
8. Binnie, M. *et al.* Long-term Outcomes of Lung Transplant With Ex Vivo Lung Perfusion. **154**, 1143–1150 (2020).
9. Buchko, M. T. *et al.* Continuous Hemodialysis Does Not Improve Graft Function During Ex Vivo Lung Perfusion Over 24 Hours. *Transplant. Proc.* **7**, 1–7 (2019).
10. Buchko, M. T. *et al.* Total parenteral nutrition in ex vivo lung perfusion: Addressing metabolism improves both inflammation and oxygenation. *Am. J. Transplant.* **19**, 3390–3397 (2019).

REVIEWERS' COMMENTS

Reviewer #1 (Remarks to the Author):

I thank to the authors for their kind answers and explanations.

Congratulations

Reviewer #3 (Remarks to the Author):

The authors have addressed my comments satisfactorily. They are now able to report the 12 month survival which has been included.